# A selective effect of dopamine on information-seeking

Valentina Vellani[1,2]*, Lianne P de Vries[1], Anne Gaule[1], Tali Sharot[1,2]*

[1]Affective Brain Lab, Department of Experimental Psychology, University College London, London, United Kingdom; [2]The Max Planck UCL Centre for Computational Psychiatry and Ageing Research, University College London, London, United Kingdom

**Abstract** Humans are motivated to seek information from their environment. How the brain motivates this behavior is unknown. One speculation is that the brain employs neuromodulatory systems implicated in primary reward-seeking, in particular dopamine, to instruct information-seeking. However, there has been no causal test for the role of dopamine in information-seeking. Here, we show that administration of a drug that enhances dopamine function (dihydroxy-L-phenylalanine; L-DOPA) reduces the impact of valence on information-seeking. Specifically, while participants under Placebo sought more information about potential gains than losses, under L-DOPA this difference was not observed. The results provide new insight into the neurobiology of information-seeking and generates the prediction that abnormal dopaminergic function (such as in Parkinson's disease) will result in valence-dependent changes to information-seeking.

## Introduction

Curiosity, commonly defined as the desire for knowledge, is a fundamental part of human nature (*Kidd and Hayden, 2015*; *Loewenstein, 1994*). In humans, it manifests as information-seeking behaviors such as asking questions, reading, conducting experiments, and online searches. Such behavior is integral to learning, social engagement, and decision-making (*Kidd and Hayden, 2015*; *Loewenstein, 1994*; *Sakaki et al., 2018*). Despite information-seeking being central to behavior, we know remarkably little about the biological mechanisms that control it.

It has been suggested that information-seeking relies on the same neural system as reward-seeking (*Bromberg-Martin and Hikosaka, 2009*; *Bromberg-Martin and Hikosaka, 2011*; *Blanchard et al., 2015*; *Charpentier et al., 2018*; *Ligneul et al., 2018*; *Kobayashi and Hsu, 2019*; *Kang et al., 2009*; *Smith et al., 2016*; *Tricomi and Fiez, 2012*; *Jessup and O'Doherty, 2014*; *Gruber et al., 2014*; *van Lieshout et al., 2018*), implying that the opportunity to gain knowledge has intrinsic value (*Grant et al., 1998*). This assumption is supported by correlational studies showing that the opportunity to gain information is encoded in regions rich in dopaminergic neurons (e.g. Ventral Tegmental Area, Substantia Nigra) and their targets (e.g. Nucleus Accumbens, Orbital Frontal Cortex) (*Bromberg-Martin and Hikosaka, 2009*; *Bromberg-Martin and Hikosaka, 2011*; *Blanchard et al., 2015*; *Charpentier et al., 2018*; *Ligneul et al., 2018*; *Kobayashi and Hsu, 2019*; *Kang et al., 2009*; *Smith et al., 2016*; *Tricomi and Fiez, 2012*; *Jessup and O'Doherty, 2014*; *Gruber et al., 2014*; *van Lieshout et al., 2018*). For example, information prediction error signals have been identified in dopamine-rich brain regions (*Bromberg-Martin and Hikosaka, 2009*), which analogous to reward prediction errors (*Schultz et al., 1997*) are theorized to provide reinforcement for seeking-information. These signals have been observed even when information is non-instrumental (*Bromberg-Martin and Hikosaka, 2009*) (i.e. cannot be used to gain future rewards or avoid future harm), consistent with the idea that the brain treats the opportunity to gain knowledge as a higher order reward (*Bromberg-Martin and Hikosaka, 2009*; *Bromberg-Martin and Hikosaka,*

**\*For correspondence:**
vellaniuni@gmail.com (VV);
t.sharot@ucl.ac.uk (TS)

**Competing interests:** The authors declare that no competing interests exist.

2011; *Blanchard et al., 2015*; *Grant et al., 1998*). Such coding may be adaptive because information could turn out to be useful in the future even if it appears useless at present (*Eliaz and Schotter, 2007*).

Thus, one hypothesis is that dopamine boosts information-seeking. However, another possibility is that dopamine selectively affects the impact of valence on information-seeking. In particular, it has been shown that individuals seek information more when information is about future gains than losses (*Charpentier et al., 2018*; *Thornton, 2008*; *Persoskie et al., 2014*; *Dwyer et al., 2015*; *Caplin and Leahy, 2001*; *Kőszegi, 2010*; *Golman et al., 2017*). For example, investors monitor their portfolio more frequently when they expect their worth has gone up rather than down (*Karlsson et al., 2009*); some people refuse to receive results of medical tests for fear of bad news (*Hertwig and Engel, 2016*); and monkeys prefer to know in advance the size of rewards they are about to receive particularly when they expect large rewards (*Bromberg-Martin and Hikosaka, 2009*; *Bromberg-Martin and Hikosaka, 2011*; *Blanchard et al., 2015*). In humans, dopaminergic midbrain regions have been shown to code for the opportunity to receive information in a valence-dependent manner (*Charpentier et al., 2018*), suggesting that the intrinsic utility of knowledge is modulated by valence.

To test the above competing hypotheses, we enhanced dopamine function in humans by administrating L-DOPA and asked them to perform an information-seeking task (*Charpentier et al., 2018*). We compared their performance to participants who received Placebo to examine whether and how dopamine alters non-instrumental information-seeking.

## Results

Two hundred and forty-eight participants performed an information-seeking task adapted from our previous publication (*Charpentier et al., 2018*), in which 16 participants did not complete the task in full; therefore, data of 232 subjects was analyzed. The study was a double-blind pharmacological intervention where one group of participants received Placebo (n = 116, females = 72, mean age = 24.36, *Table 1*) and the other received L-DOPA (150 mg) (n = 116, females = 71, mean age = 25.44, *Table 1*).

Participants began the task 40 min after receiving L-DOPA or Placebo (as in *Guitart-Masip et al., 2012*; *Sharot et al., 2009*; *Sharot et al., 2012*), as the half-life of L-DOPA is 90 min. They were endowed with £5 at the beginning of each of the four blocks to invest in two of five stocks in a simulated stock market. There were 50 trials per block. On each trial, participants observed the evolution of the market (i.e. whether the market was going up or down) and the exact value of the market (*Figure 1*). They then bid for a chance to know (or remain ignorant about) the value of their portfolio. Specifically, they indicated how much they were willing to pay to receive or avoid information about the value of their portfolio on a scale ranging from 99 p to gain knowledge through 0 p (no preference) to 99 p to remain ignorant. The more they were willing to pay, the more likely their choice was to be honored. Information was non-instrumental; it could not be used to increase rewards, avoid losses, or make changes to portfolio.

**Table 1.** Demographics.

| *Demographics* | Placebo mean (SD) | L-DOPA mean (SD) | p-Value |
|---|---|---|---|
| Age (years) | 24.36 (7.91) | 25.44 (7.92) | 0.301 |
| Gender | Females N= 72 | Females N= 71 | 0.893 |
| Income (1-9) | 4.85 (2.38) | 4.61 (2.54) | 0.462 |
| Education Level (1-10) | 7.09 (1.72) | 7.39 (1.50) | 0.157 |

There were no differences between groups in terms of demographics. p-Value is of independent sample t-test , or in the case of gender of $X^2$. Education was measured on a scale ranging from 1 (no formal education) to 10 (Doctoral degree ). Annual household income was measured on a scale from 1 (less than 10K) to 10 (more than 100K).

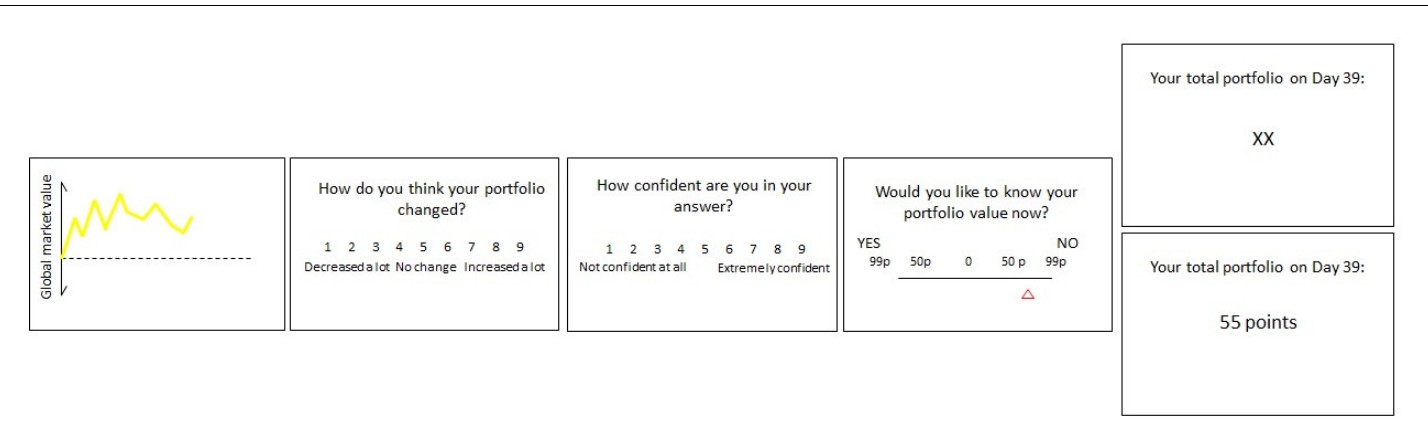

**Figure 1.** Stock market task. (**A**) Participants observed the evolution of a financial market after investing in two of its five companies. They then indicated whether they believed their portfolio value likely went up or down relative to the previous trial and indicated their confidence in their answer. They then indicated how much they were willing to pay to receive or avoid information about their portfolio value. Next, their portfolio value in points was presented on screen or hidden ('XX points' was shown).

## L-DOPA did not alter general information-seeking

L-DOPA administration did not alter general aspects of information-seeking (*Figure 2*). In particular, there were no difference between the Placebo and L-DOPA groups in the average number of trials in which participants selected to pay for information (Placebo = 71.16 trials, L-DOPA = 72.89 trials, t (230) = 0.226, p=0.821, independent samples t-test ), pay to avoid information (Placebo = 27.79 trials, L-DOPA = 27.97 trials, t(230) = 0.036, p=0.971), or not to pay at all (i.e. entered 0 p: Placebo = 93.86 trials, L-DOPA = 92.65 trials, t(230) = 0.145 p=0.885). There was also no difference in the average amount each group paid to receive information (Placebo = 18.18 p, L-DOPA = 15.68 p, t(228) = 0.928, p=0.355) or avoid it (Placebo = 11.34 p, L-DOPA = 9.95 p, t(223) = 0.587, p=0.558). These results suggest that dopamine does not generally alter information-seeking. Finally, there was no difference across groups in the number of trials participants missed (that is trials in which they were too slow in responding: Placebo = 7.03 trials, L-DOPA = 6.50 trials, t(230) = 0.283, p=0.777), suggesting no difference in engagement with the task.

## L-DOPA diminished the effect of valence on information-seeking

In this task, we had previously shown that despite participants wanting information both when the market was going down and when it was going up (*Charpentier et al., 2018*), information-seeking was modulated by the expected valence of the outcome (*Charpentier et al., 2018*). In particular, we had reported that participants were more likely to pay for information when the market was going up rather than down and more likely to pay to avoid information when the market was going down rather than up (*Charpentier et al., 2018*). This is because people expected to learn about gains when the market was going up and expected to learn about losses when the market was going down (*Charpentier et al., 2018*). The second factor we had reported to influence information-seeking was the absolute amount of change in the market. Participants were willing to pay more for information when there were big changes in the market. Here, we examine whether dopamine modulates these effects on information-seeking.

On each trial, we calculated the Willingness To Pay (WTP) for information. WTP is coded positively if participants indicated they wanted to receive information and negatively if they wanted to avoid information (*Charpentier et al., 2018*). We then ran a Linear Mixed Model to predict WTP on each trial from the two factors we had previously shown to impact information-seeking in this task (*Charpentier et al., 2018*): (i) valence (quantified as signed market change, which is the amount by which the market went up or down); (ii) absolute market change; as well as from (iii) group (L-DOPA or Placebo). All three factors were included as fixed and random effects, as were the interactions of group with each of the other two factors. Random and fixed intercepts were also included in the model.

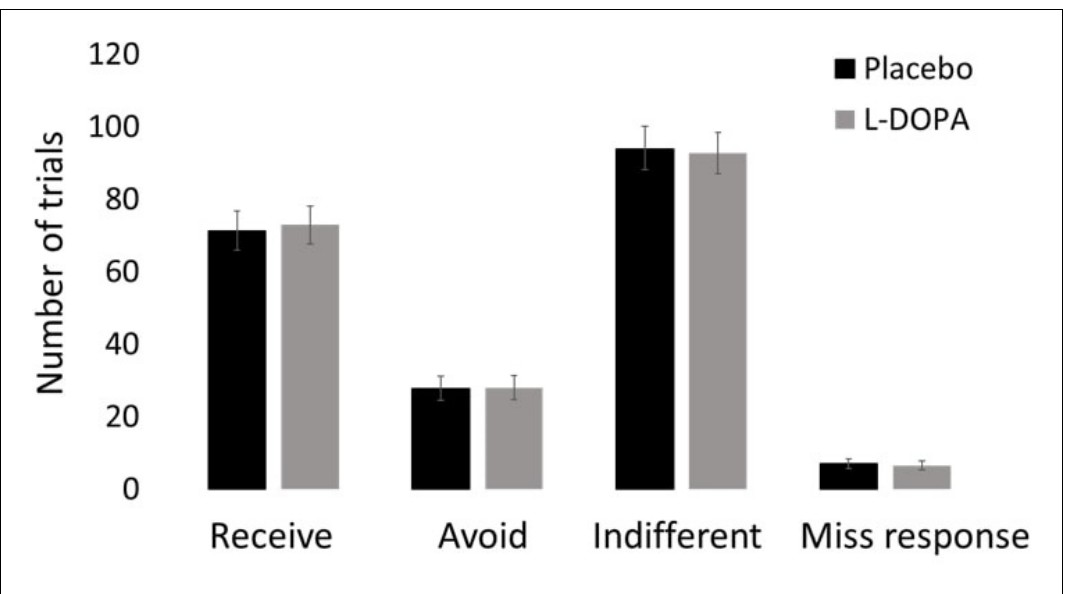

**Figure 2.** L-DOPA does not alter general aspects of information-seeking. There were no differences in general information-seeking between those who received Placebo and those who were administered L-DOPA. In particular, there were no differences in the average number of trials on which the participants decided to receive or to avoid information or were indifferent (i.e. paid 0). Furthermore, there was no difference across groups in the number of trials participants missed (that is trials in which they were too slow in responding). Error bars SEM. The online version of this article includes the following source data for figure 2:

**Source data 1.** Source data for *Figure 2*.

The results revealed an interaction between group and valence on the WTP for information (β = −0.15, CI = −0.29 /- 0.01, t(230.52) = 2.15, p=0.032) as well as a main effect of valence (β = 0.20, CI = 0.11/0.30, t(229.60) = 4.11, p=0.0001) and a main effect of absolute market change (β = 0.41, CI = 0.25/0.58, t(231.35) = 4.87, p=0.0001). There was no interaction between group and absolute market change (β = 0.07, CI = −0.17/0.30, t(232.42) = 0.576, p=0.565) nor a main effect of group (β = −2.61, CI = −7.50/2.28, t(232.53) = 1.045, p=0.297).

The interaction indicates that expected valence differentially effected the desire for information in the Placebo and L-DOPA groups. To tease apart the interaction, we next ran two mixed linear models separately for the Placebo and L-DOPA groups. WTP was entered as the dependent factor and valence and absolute market change as fixed and random factors. The model included fixed and random intercepts. This revealed a significant effect of valence in the Placebo group (main effect of signed market change: β = 0.20, CI = 0.08/0.33, t(115.25) = 3.18, p=0.001, *Figure 3a*), but lack thereof in the L-DOPA group (main effect of signed market change: β = 0.05, CI = −0.005/0.11, t(115.28) = 1.78, p=0.076, *Figure 3a*). Both groups showed a main effect of absolute market change (Placebo: β = 0.41, CI = 0.23/0.59, t(117.58) = 4.52, p=0.0001; L-DOPA: β = 0.48, CI = 0.33/0.63, t(113.54) = 6.266, p=0.0001, *Figure 3a*). These results suggest that L-DOPA selectively reduced the impact of the expected valence of information on the desire for knowledge.

The same results are observed also when using a simpler model with WTP as a dependent measure and only one independent factor - valence - coded in a binary fashion (1 for market up and 0 for market down) as fixed and random variable with fixed and random intercepts. We find a significant effect of valence in the Placebo group (β = 1.85, CI = 0.64/3.05, t(116.88)=3.01, p=0.003) with WTP for information being greater for trials in which the market went up (indicating potential gains) than down (indicating potential losses), and lack thereof in the L-DOPA group (β = 0.35, CI = −0.21/0.91, t(117.58),=1.22 p=0.224). This shows that under Placebo participants desired information more when the market was up vs down, whereas under L-DOPA the desire for information was not altered by valence.

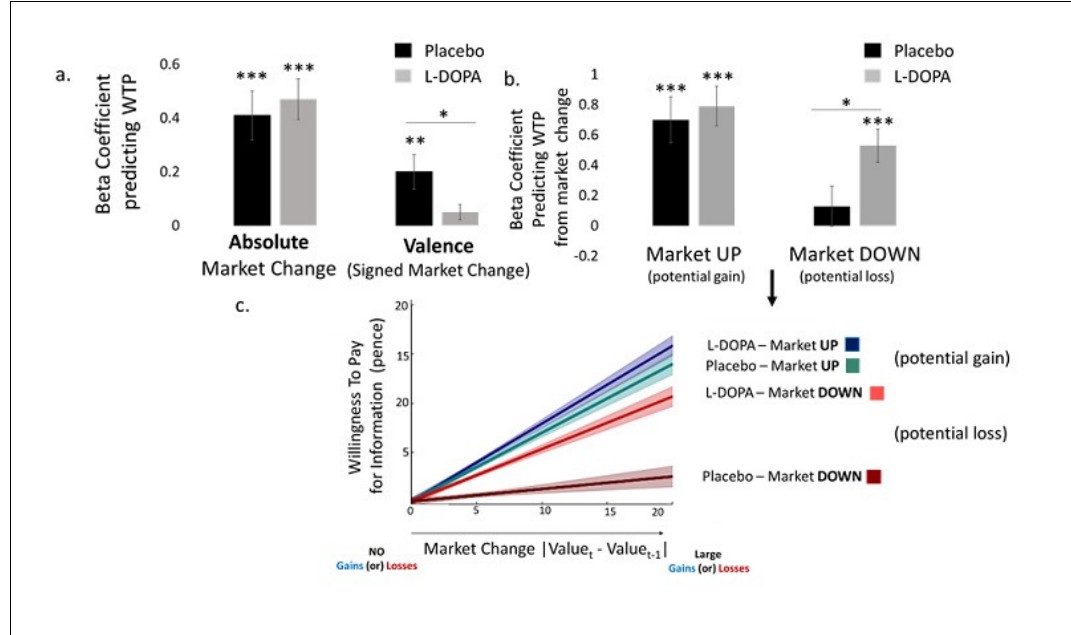

**Figure 3.** L-DOPA reduces the effect of valence on information-seeking. (a) A mixed linear regression predicting Willingness To Pay (WTP) for information revealed an interaction between group (Placebo/L-DOPA) and valence (the amount by which the market went up or down), with no interaction between group and absolute market change. To tease apart the interaction, we ran linear mixed models separately for the L-DOPA and Placebo groups. Plotted are the fixed effects of those models. As observed, this revealed a significant effect of valence on information-seeking in the Placebo group but lack thereof in the L-DOPA group. Absolute change was a significant predictor in both groups. This indicates a reduction in the influence of valence on information-seeking under L-DOPA. (b) To further characterize the effect of valence and drug on information-seeking, we run separate mixed linear models for each group and polarity predicting WTP from market change, trial number and the interaction of the two. Plotted are the fixed effects of market change for each. As can be observed under L-DOPA market change was a significant predictor of information-seeking about potential losses and gains - the greater the expected gain/loss the more participants were willing to pay for information. In contrast, under Placebo market change was a significant predictor of information-seeking about potential gains, but not losses. These results show that L-DOPA selectively alters information-seeking about losses. (c) Plotted is the effect of market change on WTP for information controlling for any effects of trial number. As can be observed the slopes are significantly positive for all groups/conditions except for the Placebo group in the loss domain. Clouds are based on Standard Errors of the fixed effect. Error bars SEM, * p <0.05, ** p < 0.01, *** p < 0.001.
The online version of this article includes the following source data for figure 3:

**Source data 1.** Source data for Figure 3.

## L-DOPA selectively alters information-seeking about potential losses

Our results indicate that L-DOPA selectively reduces the impact of valence on information-seeking. Next, we ask if this effect is due to L-DOPA altering information-seeking about potential *losses*, about potential *gains*, or both. Moreover, we ask whether the effect of L-DOPA emerged over the course of the experiment or whether it was apparent from the very beginning.

To that end, we ran two separate mixed effect linear model predicting the WTP for information – one for trials in which the market went up (potential gain trials) and one for which the market went down (potential loss trials). The independent factors included (i) market change, (ii) trial number, and (iii) group (L-DOPA/Placebo). As each model now includes only one polarity - either market going up or down - signed market change and absolute market change are perfectly correlated. Thus, only one factor 'market change' is added. In the loss domain, the greater the 'market change' the greater the expected losses. In the gain domain, the greater the 'market change' the greater the expected gains. All three factors and their interactions were included as fixed and random effects. Random and fixed intercepts were also included in the model. The results revealed an interaction between market change and group in the loss domain (β = 0.40, CI = 0.07/0.74, t (751.900)=2.35,

p=0.018), but not in the gain domain (β = 0.10, CI = −0.29/0.50,t (490.600)=0.505, p=0.614) where instead there was a main effect of market change (β = 0.70, CI = 0.42/.98, t (495.800)=4.896, p=0.0001). No other effects were significant.

To characterize the interaction of interest (between market change and group) in the loss domain and lack thereof in the gain domain, we ran four linear mixed models - one for each group and valence polarity. WTP was the dependent factor, and the independent factors were (i) market change and (ii) trial number. Both factors and their interactions were included as fixed and random effects. Random and fixed intercepts were also included in the model. This revealed that under L-DOPA participants were willing to pay more for information the greater the gains (effect of market change: β = 0.79, CI = 0.53/1.05, t (313.300)=5.98, p = 0.0001 *Figure 3b*) a*nd* the greater the losses (effect of market change: β = 0.53, CI = 0.33/0.74, t (484.700)=5.033, p=0.0001, *Figure 3b*). In contrast, under Placebo participants were willing to pay more for information the greater the gains (effect of market change: β = 0.69, CI = 0.40/0.99, t (219.92)=4.64, p=0.0001, *Figure 3b*) *but* did not show this effect for losses (effect of market change: β = 0.12, CI = −0.13/0.39, t (335.800) =0.956, p=0.33, *Figure 3b*). For L-DOPA in the gain domain, there was an additional interaction between trial and market change (β = −0.002, CI = −0.004 /- 0.006), t (382.20) = 2.74, p=0.006). No other effects were significant.

The results show that under L-DOPA, participants' desire for information increased as the expected magnitude of the outcome increased - participants were willing to pay more for information as potential gains *and* losses increased (*Figure 3C*). In contrast, under Placebo, participants' desire for information increased as potentials gains increased but remained constant and relatively low for potential losses (*Figure 3C*).

## The effect of L-DOPA on information-seeking for losses is not explained by changes in expectations

We next ask whether the selective effect of L-DOPA on information-seeking about losses can be explained by a selective effect of L-DOPA on expectations about losses. To test participants' expectations regarding their outcomes, we asked participants whether they believed their stocks went up or down after observing the global market change. This was done by having participants rate their expectations on a scale ranging from 1 (decreased a lot) to 9 (increased a lot). We then entered these ratings into Linear Mixed Model predicting expectation ratings. The independent factors were: (i) valence (signed market change), (ii) absolute market change, and (iii) group (L-DOPA or Placebo). All three factors were included as fixed and random effects as were the interactions of group with each of the other two factors. Random and fixed intercepts were also included in the model. There was no main effect of group (β = 0.06, CI = −0.05/0.17, t (257.400)=1.01, p=0.310), nor an interaction between group and valence (β = −0.01, CI = −0.03/0.02, t (232.900)=0.63, p=0.529) nor an interaction between group and absolute market change (β = 0.00, CI = −0.01/0.01, t (242.900) =0.266, p=0.790). There was a main effect of absolute market change (β = −0.01, CI = −0.02 /- 0.0001, t (241.000)=2.28, p=0.023) and of valence (β = 0.21, CI = 0.19/0.22, t (233.000)=23.264, p=0.0001). The latter confirms that participants' expectations about their outcomes were linked to the observed trends in the market.

These results suggest that L-DOPA did not affect participants' expectations. To further examine whether there may be an effect of L-DOPA on expectations that altered over time, we added to the model above trial number as a fixed and random factor as well as all the two- and three-way interactions of trial number with the other factors. Once again, this neither revealed an effect of group on expectations (β = 0.05, CI = −0.49/0.60, t(312.800) = 0.186 p=0.852) nor were any of the interactions between group and any of the other factors significant (all Ps > 0.329). These results suggest that L-DOPA selectively altered the effect of valence on information-seeking without altering outcome expectations.

## Discussion

Humans and non-human animals seek information even when information cannot be used to alter outcomes (*Bromberg-Martin and Hikosaka, 2009*; *Bromberg-Martin and Hikosaka, 2011*; *Blanchard et al., 2015*; *Charpentier et al., 2018*). This observation led to the notion that knowledge may have evolved to carry intrinsic value (*Bromberg-Martin and Hikosaka, 2009*; *Bromberg-*

*Martin and Hikosaka, 2011*; *Blanchard et al., 2015*; *Grant et al., 1998*). Indeed, it has been shown that the opportunity to receive non-instrumental information is encoded by the same neural system as for primary rewards (*Bromberg-Martin and Hikosaka, 2009*; *Bromberg-Martin and Hikosaka, 2011*; *Blanchard et al., 2015*; *Charpentier et al., 2018*; *Ligneul et al., 2018*; *Kang et al., 2009*; *Gruber et al., 2014*; *van Lieshout et al., 2018*). As this system includes regions rich in dopamine (the findings triggered the hypothesis that dopamine plays a critical role in non-instrumental information-seeking (*Bromberg-Martin and Hikosaka, 2009*; *Bromberg-Martin and Hikosaka, 2011*; *Blanchard et al., 2015*; *Charpentier et al., 2018*). By manipulating the dopamine levels in humans, we were able to directly test this hypothesis.

Our results show that L-DOPA has a selective effect on non-instrumental information-seeking. Administration of L-DOPA dampened the effect of valence on non-instrumental information-seeking, altering non-instrumental information-seeking about potential losses without impacting non-instrumental information-seeking about potential gains. Specifically, while participants under Placebo sought information more about potential gains than losses (an effect observed in the past [*Charpentier et al., 2018*]), under L-DOPA this difference was not observed. Moreover, under L-DOPA, participants' WTP for information increased as potential gains *and* losses increased. In stark contrast, under Placebo, participants' WTP for information increased as potential gains increased but remained constant and relatively low as potential losses increased.

An intriguing question concerns the mechanism by which L-DOPA alters information-seeking about potential losses. The effect could not be explained by changes to participants' mood, as there were no differences in participants' self-reported subjective state under Placebo and L-DOPA (see *Table 2*). Neither could it be explained by reduced attention and/or engagement, as participants under L-DOPA did not miss more trials than those under Placebo. L-DOPA also did not alter expectations of outcomes. Thus, modulation of outcome expectations (that is how much is expected to be lost/gained) cannot explain the results. Moreover, as the task did not involve learning (past outcomes had no impact on future outcomes, see supplementary results), L-DOPA did not affect learning about potentials outcome gains and losses.

**Table 2.** Subjective State Questionnaire.
Subjective State Questionnaire (*Joint Formulary Committee, 2009*) revealed no differences in subjective state between groups. p-Value relates to independent sample t-test.

| *Subjective State Questionnaire* | Before the task | | | After the task | | |
|---|---|---|---|---|---|---|
| | Placebo mean (SD) | L-DOPA mean (SD) | p-Value | Placebo mean (SD) | L-DOPA mean (SD) | p-Value |
| Alert to drowsy | 2.68 (1.19) | 2.62 (1.08) | 0.687 | 3.60 (1.41) | 3.85 (1.57) | 0.208 |
| Calm to excited | 2.33 (1.11) | 2.29 (1.03) | 0.808 | 2.34 (1.09) | 2.27 (1.22) | 0.632 |
| Strong to feeble | 2.68 (1.01) | 2.63 (1.01) | 0.699 | 2.97 (1.13) | 3.15 (1.35) | 0.264 |
| Muzzy to clear headed | 4.47 (1.26) | 4.48 (1.11) | 0.956 | 3.70 (1.23) | 3.41 (1.39) | 0.099 |
| Coordinated to clumsy | 2.28 (1.14) | 2.22 (1.06) | 0.722 | 2.80 (1.16) | 3.02 (1.31) | 0.187 |
| Lethargic to energetic | 3.89 (1.14) | 3.94 (1.18) | 0.736 | 3.20 (1.24) | 3.00 (1.43) | 0.263 |
| Contented to discontented | 2.18 (1.01) | 2.12 (0.83) | 0.620 | 2.58 (1.16) | 2.65 (1.19) | 0.644 |
| Troubled to tranquil | 4.83 (1.02) | 4.66 (1.02) | 0.201 | 4.52 (1.13) | 4.55 (1.13) | 0.858 |
| Slow to quick witted | 4.32 (1.11) | 4.28 (1.04) | 0.761 | 3.63 (1.30) | 3.31 (1.30) | 0.069 |
| Tense to relaxed | 4.64 (1.11) | 4.67 (0.98) | 0.803 | 4.47 (1.14) | 4.42 (1.23) | 0.733 |
| Attentive to dreamy | 2.78 (1.26) | 2.73 (1.10) | 0.740 | 3.47 (1.34) | 3.42 (1.38) | 0.804 |
| Incompetent to proficient | 4.56 (0.98) | 4.70 (0.86) | 0.260 | 4.14 (1.16) | 4.02 (1.26) | 0.450 |
| Happy to sad | 2.43 (1.06) | 2.34 (0.84) | 0.453 | 2.62 (1.12) | 2.57 (0.95) | 0.712 |
| Antagonistic to friendly | 5.08 (0.97) | 5.07 (0.81) | 0.942 | 4.65 (0.95) | 4.60 (1.04) | 0.703 |
| Interested to bored | 2.35 (1.21) | 2.28 (1.02) | 0.639 | 3.52 (1.45) | 3.63 (1.50) | 0.587 |
| Withdrawn to sociable | 4.34 (1.17) | 4.36 (1.18) | 0.868 | 3.90 (1.17) | 3.83 (1.39) | 0.672 |

One possibility is that L-DOPA altered expectations not about outcomes per-se, but about the affective impact of negative information. A negative cue (e.g. watching the financial market fall) triggers expectations not only about the material outcome (the amount one has likely lost) but also about how bad it would be to receive information about that loss (*Bromberg-Martin and Sharot, 2020*). L-DOPA may have triggered less pessimistic expectations regarding the latter, altering the value of information about losses, which could have changed information-seeking in the loss domain. To illustrate this point, imagine two participants who accurately expect to lose £100 when they observe the market falling. One participant predicts that learning about the loss will have little negative impact, whereas the other predicts a large negative impact. Dopamine dips could signal both elements separately when observing the cue. As L-DOPA is thought to interfere with such dips (*Ungless et al., 2004*; *Satoh et al., 2003*), it could result in less pessimistic expectations about the value of bad news and thus more information-seeking. This possibility can be investigated in the future by recording participant's actual and predicted expectations regarding the affective impact of information.

It is important to keep in mind that our task exclusively examined non-instrumental information about gains and losses. As dopamine is known to play an important role in reward-guided learning and decision-making, it is possible that dopamine plays a more general role in information-seeking when information has instrumental value and/or for non-valenced information. Future studies are needed to investigate the role of dopamine in those situations.

Because information-seeking is integral to decision-making (*Kidd and Hayden, 2015*; *Loewenstein, 1994*), understanding its biological basis is important for understanding impairments in these domains. Our results suggest that patients with deficiency to the dopamine system may exhibit abnormal patterns of information-seeking, which may provide a marker of their condition. For example, patients with low levels of dopamine function, such as patients with Parkinson's disease, may be less likely to seek information regarding negative events. The findings also generate predictions of how prescription drugs targeting dopamine function may alter patients' information-seeking behavior. For example, patients taking L-DOPA may increase self-exposure to negative information, which may induce negative affect.

## Materials and methods

**Key resources table**

| Reagent type (species) or resource | Designation | Source or reference | Identifiers | Additional information |
|---|---|---|---|---|
| Software, algorithm | SPSS | SPSS | RRID:SCR 002865 | Version 25 |
| Software, algorithm | MATLAB | MATLAB | RRID:SCR_001622 | Version R2020a |
| Software, algorithm | R | R | RRID:SCR_001905 | R-4.0.1 |
| Chemical compound, drug | levodopa | Orion Pharma (UK) Limited | PubChem CID:6047 | 150 mg |
| Chemical compound, drug | carbidopa | Orion Pharma (UK) Limited | PubChem CID: 34359 | 37.5 mg |
| Chemical compound, drug | entacapone | Orion Pharma (UK) Limited | PubChem CID: 5281081 | 200 mg |

### Participants

Two hundred and forty-eight subjects were recruited via the University College London psychology online system and assigned randomly to receive Placebo (123) or L-DOPA (125). Sample size was calculated based on our previous studies (*Sharot et al., 2009*; *Sharot et al., 2012*) looking at dopamine effects on decision-making. All participants filled in the informed consent and a screening form for significant medical conditions, medications, and illicit drugs. All subjects were paid for their participation. The study was double-blind and approved by the UCL ethics committee (Project ID Number: 8127/001).

Data from five subjects was lost due to technical error, and 11 subjects did not complete the task due to either feeling nausea (five subjects), power outage (one subject) or lack of interest/motivation

(five subjects). Thus, we obtained full data sets from 232 participants (Placebo group: n = 116, females = 72, mean age = 24.36, SD = 7.918; L-DOPA group: n = 116, females = 71, mean age = 25.44, SD = 7.926). Education level was measured on a scale from 1 (no formal educatio) to 10 (Doctoral Degree). Income was measured on a scale from 1 (annual household income £10,000 or less), to 9 (annual household income over £100000). There were no significant differences between the groups in terms of age (t(230) = 1.036, p=0.301), income (t(228) = 0.737, p=0.462), gender ($X^2$(1) = 0.018, p=0.893), and education level (t(230) = 1.420, p=0.157).

## Procedure and task

Participants were administered either Placebo or L-DOPA (150 mg of levodopa, 37.5 mg of carbidopa, and 200 mg of entacapone) upon arrival to the lab in a double-blind fashion. They then completed a brief questionnaire - the Subjective State Questionnaire (SSQ) (*Joint Formulary Committee, 2009*). They began the task 40 min after the administration of L-DOPA/Placebo (L-DOPA half-life is 90 min and peaks at 60 min). The task took about 60 min to complete after which they completed the SSQ (*Joint Formulary Committee, 2009*) again. There was no differences between the Placebo and L-DOPA groups across SSQ (*Joint Formulary Committee, 2009*) items either before or after the task (see *Table 2*).

The task, known as the Stock Market Task, was adapted from our previous study (*Charpentier et al., 2018*). This task is composed of four blocks of 50 trials each. At the beginning of each block, each participant received 50 points, worth £5, which they had to invest in 2 of 5 five fictitious companies which compose a 'global market'. On each trial, participants first observed changes in market value (a dynamic increase or decrease in the curve lasting 2.3 s). The market value fluctuations reflected changes in the overall market; therefore, it partially indicated changes in the participant's own portfolio value. Unbeknown to the participants, on each trial, there was a 65% probability that their actual portfolio value would change consistent with the market trend. After observing the global market change, participants were asked to predict how their portfolio value likely changed relative to the previous trial from 1 (decreased a lot) to 9 (increased a lot) and their confidence in their answer from 1 (not confident at all) to 9 (extremely confident). They had up to 8 s to perform each rating. Sixty-four subjects (34 subjects received Placebo and 30 L-DOPA) were asked to state their expectation and confidence on their answer only on blocks 3 and 4, while all other subjects were asked to respond on every trial.

Participants were then given the chance to discover their portfolio value on that trial. Subjects had up to 8 s to state how much they were willing to pay to either receive or avoid information about their portfolio value. They could state their decision using a scale ranging from 99 p to avoid information ('NO'), through 0, to 99 p to receive information ('YES') (p indicated pence). Position of 'YES' and 'NO' (left/right) were counterbalanced across participants. They were informed that the more they paid the greater the probability that their wish would be honored. When 0 p was selected, information was delivered 50% of the time. If they selected an amount between 1 p and 20 p, their request was honored on 55% of the trials, between 21 p and 40 p - 65%, and so on up to 95%. Participants were not aware of these exact mathematical relationships. After that, the current value of their portfolio was shown on screen or hidden (that is 'XX points' was shown) for 3 s. In this study, information was not instrumental, in the sense that it could not be used to change the portfolio.

At the end of the task, one trial was randomly selected and participants received the value of their portfolio on that trial (e.g, portfolio value of 60 points=£6). If on that trial they decided to pay a certain amount to receive or avoid information and their wish was honored (e.g. they paid 40 p to receive information and they received it), then that amount was deducted from the portfolio value (e.g. £6-£0.40 = £5.60).

## Data analysis

First, we investigated the effect of dopamine manipulation on general aspects of information-seeking by comparing the number of trials in which subjects decided to pay to receive information, avoid information, or pay nothing, the average amount they paid to receive information, the average amount they paid to avoid information and number of missed trials between the L-DOPA and the Placebo groups with an independent samples t-test.

Then, we computed willingness to pay (WTP) on every trial with amount paid to avoid information scored negatively, and amount paid to receive information positively (zero is simply coded as zero). For each trial, a Linear Mixed Model was run to predict WTP from the two factors we had previously shown to impact information-seeking in this task (*Charpentier et al., 2018*) (i) valence (quantified as signed market change, which is the amount by which the market went up or down); (ii) absolute market change; as well as from (iii) group (L-DOPA or Placebo). All three factors were included as fixed and random effects, as were the interactions of group with each of the other two factors. Random and fixed intercepts were also included in the model. All linear mixed models were run in R using the lmer function (lme4 package) using maximum likelihood estimation method, the BOBYQA (Bound Optimization BY Quadratic Approximation) optimizer and a maximum number of iterations of 100,000.

As the model revealed a group by valence interaction, we next ran two mixed linear models separately for the Placebo and L-DOPA groups to tease apart that interaction. WTP was entered as the dependent factor and valence and absolute market change as fixed and random factors. The model included fixed and random intercepts. We also ran simpler models for each group separately, with WTP as a dependent measure and valence, coded in a binary fashion (market up/down), as fixed and random variable with fixed and random intercepts.

As the above analysis revealed a significant effect of valence in the Placebo group but not the L-DOPA group, we asked if the effect is due to L-DOPA altering information-seeking about potential losses, about potential gains, or both. Moreover, we ask whether the effect of L-DOPA emerged over the course of the experiment or whether it was apparent from the very beginning. Thus, we ran two separate mixed effect linear model predicting the WTP for information – one for trials in which the market went up (potential gain trials) and one for which the market went down (potential loss trials). The independent factors included (i) market change, (ii) trial number (iii), and group (L-DOPA/Placebo). As each model now includes only one polarity - either market going up or down - signed market change and absolute market change are perfectly correlated. Thus, only one factor 'market change' is added. In the loss domain, the greater the 'market change' the greater the expected losses. In the gain domain, the greater the 'market change' the greater the expected gains. All three factors and their interactions were included as fixed and random effects. Random and fixed intercepts were also included in the model. We followed up with four linear mixed models - one for each group and valence polarity. WTP was the dependent factor and the independent factors were (i) market change (ii) and trial number. Both factors and their interactions were included as fixed and random effects. Random and fixed intercepts were also included in the model.

Finally, we examined whether participants' expectations are affected by L-DOPA. To this aim, we run a Linear Mixed Model predicting expectations with the following independent factors: (i) valence (signed market change), (ii) absolute market change, and (iii) group (L-DOPA or Placebo). All three factors were included as fixed and random effects as were the interactions of group with each of the other two factors. Random and fixed intercepts were also included in the model. To further examine whether there may be an effect of L-DOPA on expectations that alters over time, we added to the model above trial number as a fixed and random factor as well as all the two- and three-way interactions of trial number with the other factors.

## Data availability

Anonymized data are available on GitHub (https://github.com/affective-brain-lab/A-Selective-Effect-of-Dopamine-on-Information-Seeking-Valentina-Vellani-; *Vellani, 2020* copy archived at swh:1:rev:71ef6f1b6a438236450207810b0630c8738336b8).

## Acknowledgements

The research was funded by a Wellcome Trust Senior Research Fellowship 214268/Z/18/Z to TS. We thank Lili Lantos and Sims Witherspoon for assistance in collecting data; Rick Adams for providing medical support; Bastian Blain, Irene Cogliati Dezza, Laura Katharina Globig and Christopher Kelly for providing comments on a previous version of the manuscript. This study has been approved by the UCL Research Ethics Committee (Project ID Number: 8127/001).

## Additional information

### Funding

| Funder | Grant reference number | Author |
|---|---|---|
| Wellcome Trust | Wellcome Trust Senior Research Fellowship 214268/Z/18/Z | Tali Sharot |

The funders had no role in study design, data collection and interpretation, or the decision to submit the work for publication.

### Author contributions

Valentina Vellani, Data curation, Formal analysis, Investigation, Visualization, Writing - original draft, Project administration; Lianne P de Vries, Data curation, Formal analysis, Investigation, Writing - review and editing; Anne Gaule, Data curation, Investigation, Writing - review and editing; Tali Sharot, Conceptualization, Resources, Formal analysis, Supervision, Funding acquisition, Visualization, Methodology, Writing - original draft, Project administration

### Author ORCIDs

Valentina Vellani (iD) https://orcid.org/0000-0001-7191-5257
Lianne P de Vries (iD) https://orcid.org/0000-0002-4643-6200
Tali Sharot (iD) https://orcid.org/0000-0002-8384-6292

### Ethics

Human subjects: informed consent was given by all subjects. The study was approved by the departmental ethics committee at UCL (Project ID Number: 8127/001).

### Decision letter and Author response

Decision letter https://doi.org/10.7554/eLife.59152.sa1
Author response https://doi.org/10.7554/eLife.59152.sa2

## Additional files

### Supplementary files

• Transparent reporting form

### Data availability

Anonymized data is available on GitHub (https://github.com/affective-brain-lab/A-Selective-Effect-of-Dopamine-on-Information-Seeking-Valentina-Vellani-) (copy archived at https://archive.software-heritage.org/swh:1:rev:71ef6f1b6a438236450207810b0630c8738336b8/).

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

# Appendix 1

## Supplementary material

As described in the method section, our task was not designed to be a learning task. Rather, the task was a non-instrumental task where subjects could not influence outcomes. Neither were they incentivized to generate accurate expectations regarding outcomes. Nor did past outcomes have any bearing on present outcomes. The likelihood that outcomes (that is change to portfolio value) will follow the same trend as the market was 65% and in 35% it would change in the opposite direction with a randomly generated magnitude. Thus, the most accurate way to make a prediction is simply to rely on the market change on the present trial regardless of previous outcomes. Indeed, we have previously shown that participants are unaffected by trial history when making predictions on present trials in this task (*Charpentier et al., 2018*).

Nevertheless, we tested whether there were any effects of past trials on expectations on present trials regrading portfolio outcomes. In particular, we run a mixed linear model predicting participants' expectation rating on trial t (note that the rating is always about change in portfolio on that trial relative to previous trial) from past outcome (portfolio t-1). They were not (L-DOPA group: $\beta$ = 0.0001, CI = -0.003/0.002, t(519.000) = 0.464, p = 0.643; Placebo group: $\beta$ = 0.0009, CI = -0.001/0.003, t(213.500) = 0.621, p = 0.535). We also tested whether current expectations were related to the difference between change in market on last trial (portfolio t-1 minus portfolio t-2) and expectation rating on last trial. In this analysis, we only included trials in which portfolio value was observed on the last trial. They were not (L- DOPA: $\beta$ = -0.001, CI = -0.01/0.008, t(120.787) = 0.315, p = 0.753; Placebo: $\beta$ = 0.001, CI = -0.009/0.012, t(108.300) = 0.304, p = 0.762). As participants often did not observe the portfolio value on trial t-2 we ran the analysis again this time instead of inserting portfolio t-2 in the equation above we inserted the portfolio value last observed before t-1. Again, this did not predict subjects' expectations (L-DOPA: $\beta$ = -0.001, CI = -0.008/0.006, t(112.877) = 0.330, p = 0.742; Placebo: $\beta$ = -0.0007, CI = -0.006/0.005, t(102.700) = 0.253, p = 0.800). We then examined whether wiliness to pay for information on the current trial was influenced by previous outcomes by running all these models again, this time predicting WTP for information. As expected, none showed a significant effect (all P > 0.240). This analysis confirms that subjects did not treat this task as an outcome learning task.

## Indifferent trials

To examine if L-DOPA and valence altered the number of trials in which participants decided to pay 0p ('indifferent trials') we conducted a repeated measures ANOVA with group (L-DOPA/Placebo) as a between subject variable and valence (market up/down) as a within subject variable. There was not an effect of valence (F(1,230) = 3.025, p = 0.083) nor an effect of group (F(1,230) = 0.021, p = 0.885) or an interaction (F(1,230) = 0.080, p = 0.778). There were no differences between groups regarding the number of indifferent trails when the market went up (t(230) = 0.175 p = 0.861) or down ( t(230) = 0.112 p = 0.911). Note, that indifferent trials are included in all the analysis in the main text.

