## [Decision Letter]

**Acceptance summary:**

Your investigation of the role of dopamine in human information seeking using a well-powered pharmacological intervention has provided novel observations for existing theories. The finding that L-DOPA administration in healthy volunteers reduces the impact of valence on information seeking in a stock market task makes testable predictions regarding alterations of information seeking in patients suffering from abnormal dopaminergic function. Congratulations for this insightful study.

**Decision letter after peer review:**

Thank you for submitting your article "A selective effect of dopamine on information-seeking" for consideration by *eLife*. Your article has been reviewed by three peer reviewers, including Valentin Wyart as the Reviewing Editor and Reviewer #1, and the evaluation has been overseen by Christian Büchel as the Senior Editor.

The reviewers have discussed the reviews with one another and the Reviewing Editor has drafted this decision to help you prepare a revised submission.

This manuscript describes a human pharmacology study which aims at characterizing the role of dopamine in information seeking. The authors contrast two hypotheses: 1) a general increase in information seeking, and 2) a selective increase in information seeking about potential losses. Using a stock “market task” tested on two groups of subjects (N = 116 each) under placebo and L-DOPA, they provide support for the second hypothesis. They report that subjects under placebo seek information about potential gains more than potential losses (as previously reported), and that L-DOPA reduces this asymmetry by increasing information seeking about potential losses.

The research question is of interest to a large research community interested in the substrates of information seeking under uncertainty. The use of a pharmacological protocol tested on a large group of subjects appears adequate to study the research question, and the reported results could provide new insights on information seeking. However, despite these merits, the reviewers have raised concerns regarding the analyses conducted to obtain some of the key results – concerns that the authors should address in a revised version of the manuscript. The outcome of the additional analyses used to address the reviewers' concerns will be critical for us to decide whether the manuscript can be published at *eLife*. The paragraphs below describe the main concerns that have been discussed among reviewers, and that should be addressed explicitly in a revised version of the manuscript. The separate reviews from the three reviewers are attached at the bottom of this message for your reference, but they do not require point-by-point responses.

Main concerns that require revisions:

1) Status of information seeking in the task. As emphasized by the authors in the description of their task, information seeking in their “stock market” task is purely non-instrumental. It cannot be leveraged by subjects to gain information that would help them maximizing rewards in the task – something which subjects should be aware of from the task instructions. We agree that information seeking behaviors can be studied in situations where it has no instrumental value. However, it is unclear how one can assume that findings obtained in a condition where information seeking is divorced from reward maximization would generalize to conditions where information seeking has an instrumental value. This is important in this study because the authors study the effect of dopamine on information seeking, and dopamine is also widely known to play an important role in reward-guided learning and decision-making. It is thus important to state, and discuss, the possibility that the results obtained in this study may not apply to tasks where information seeking can be leveraged to maximize rewards.

2) Handling of “indifferent” trials and exclusion of subjects. It is unclear to me how the authors dealt with the large number of “indifferent” information seeking ratings provided by the subjects (this is the most populated class of trials in Figure 2). The authors should clarify whether and how this large number of trials was handled in the analyses. We recommend reporting whether the groups differed in the number of trials they were indifferent under the same conditions (e.g., market-up vs. market-down) – ideally the proportion of each of these 3 choice types would be presented at least in a supplementary figure. Also, the exclusion of participants based on the empirical variability of their choices appears suboptimal. The authors have adopted a two-step approach when analyzing willingness to pay and choice – each participant's betas from individual regressions were entered into t-tests against zero, as well as independent-sample t-tests for comparisons between groups. We recommend the authors instead run a linear mixed effects model, including interactions with treatment group alongside the effects of the market variables. The benefit of this approach would be that all participants can be included and the number of tests to run would be minimized – thus reducing the type-1 error rate. Further, due to the hierarchical fitting and shrinkage, the obtained parameters would be more robust. As an example, for the analysis of the choice data, a number of subjects needed to be excluded because they did not have sufficient observations in either level of the DV. For these data, we recommend the authors instead use a generalized linear mixed effects model with a binomial link function to replace the logistic regression.

3) Fixed-effects analyses in linear mixed models. The reliance on fixed-effects estimation of interaction terms rather than random-effects estimation seems problematic, given that all the key results in the manuscript depend on interaction terms. It appears that the authors did not fit the data for every participant separately but instead just fit the ratio of participant choices across trials separately for each group. If this is what they did, it is quite unusual and there is no rationale for it mentioned in the text. More commonly the model would have been fit on the individual participant level – preferably using a hierarchical approach. It is also completely unclear why they suddenly introduce this type of modeling at this stage. If these analysis choices have a clear rationale, they should be clarified explicitly in the main text since the key results depend on it. However, by aggregating behavioral measures within groups and trials and market condition, and then analyzing the data with group as intercept and trial and the market condition as predictors, the authors are effectively using a fixed-effects approach and thus do not account for between-participant variability. In practice, the authors are generating two artificial participants (placebo and L-dopa) and then assessing the difference between those two, without accounting for between-participant variability. We recommend the authors instead run a linear mixed effects model across all subjects, with subject as random intercept and group as a between-subject regressor, and test for interactions between group, trial and the predictors of interest.

4) Reward history effects to link the results more directly to DA and reward-guided learning. The authors interpret their effects in part through the lens of DA influences on encoding of positive vs. negative RPEs, but this interpretation does not rest on specific analyses of the behavioral data. Because the “stock market” task is a prediction task, it would be very useful to know better how subjects learnt to make expectations in this task under placebo and L-DOPA. For this purpose, the authors could fit a canonical RL model (e.g., with a Rescorla-Wagner rule) to estimate the learning parameters of the subjects in the two groups. They could also (or instead) look at trial history effects (prior outcomes vs. expectations) on subsequent expectations and choices in the two groups. Because the authors are proposing in the Discussion that dopamine affects predictions, it would be very valuable to model the formation of predictions (through basic RL, for example) to know whether and how dopamine (L-DOPA) influences this learning process.

5) Implications of significant time-dependent interactions. The fact that it is the three-way interaction between group, valence and time that is significant, not the two-way interaction between group and valence has implications. This result suggests that there was no maintained valence effect but instead an effect that diminishes over time. It is difficult to know whether that is because the drug effects got weaker or other reasons, but in any case it deserves more attention – and explicit discussion – as the paper currently mostly reads as if there was a straightforward valence effect without changes over time. This is of course unless the authors can provide additional evidence for a time-independent valence by group interaction.

Reviewer #1:

This manuscript describes a human pharmacology study which aims at characterizing the role of dopamine in information seeking. The authors contrast two hypotheses: 1) a general increase in information seeking, and 2) a selective increase in information seeking about potential losses. Using a stock “market task” tested on two groups of subjects (N = 116 each) under placebo and L-DOPA, they provide support for the second hypothesis. They report that subjects under placebo seek information about potential gains more than potential losses (as previously reported), and that L-DOPA reduces this asymmetry by increasing information seeking about potential losses.

The studied question is of interest to a large research community interested in the substrates of information seeking under uncertainty. The pharmacological protocol on large groups of subjects appears adequate to test the studied question, and the reported results provide new insights on information seeking. The manuscript is also clearly written and introduces the existing research appropriately. However, I have concerns regarding the status of information seeking in the tested “stock market” task, and regarding some of the analyses conducted to obtain some of the key results – concerns which the authors should address in a revised version of the manuscript.

Main concerns:

1) Status of information seeking in the task. As emphasized by the authors in the description of their task, information seeking in their “stock market” task is purely non-instrumental. It cannot be leveraged by subjects to gain information that would help them maximizing rewards in the task – something which subjects should be aware of from the task instructions. I completely understand the authors' line of reasoning that information seeking behaviors are observed even in situations where the information has no instrumental value, and thus that information seeking can be studied in their protocol. However, I don't see how one can assume that the authors' findings, obtained in a condition where information seeking is divorced from reward maximization, would extend to conditions where information seeking has an instrumental value. This is important in this study because the authors study the effect of dopamine on information seeking, and dopamine is also widely known to play an important role in reward-guided learning and decision-making. The pattern of effects observed here in a study where information seeking is non-instrumental may thus change substantially when information seeking interacts with reward maximization. I am not asking here for additional data collection, of course, but I think it would be important to discuss not only the strengths (i.e., the fact that information seeking can be studied in isolation when divorced from reward maximization), but also the limitations of the current study (e.g., that the results obtained in this study may change in a task where information seeking can be leveraged to maximize rewards).

2) Inclusion of “indifferent” trials in reported analyses. After reading the manuscript, it was unclear to me how the authors dealt with the large number of “indifferent” information seeking ratings provided by the subjects (this is the most populated class of trials in Figure 2). The authors should clarify how this large number of trials was included in the analyses. Also, the exclusion of participants based on their choice variability seems suboptimal. Could the authors use a hierarchical model to retain all participants in the analyses? It seems like the current approach (performing single-subject analyses and then averaging parameter estimates at the group level) is suboptimal compared to a hierarchical model which could include all subjects in the analysis, even those with limited choice variability.

3) Fixed-effects analyses in Linear Mixed Models. The sole reliance on fixed-effects estimation of interaction terms (all the key results depend on interaction terms) rather than random-effects estimation seems problematic. After reading the manuscript, I was unclear regarding the exact models that were used, and thus uncertain of their validity. In any case, the authors could in theory (unless they rely solely on between-subject variability to assess the interaction terms) design hierarchical mixed-effects models to describe and estimate interaction terms, which I think would be more appropriate given their design.

4) All-or-none expectation ratings, and modeling of gradual learning of expectations. Figure 4B seems to suggest that expectation ratings provided by subjects were “all-or-none” (either strongly toward “expect loss”, or strongly toward “expect gain”). Is it the case? The authors could plot the overall distribution of expectation ratings on Figure 4B next to the current plot. Because the “stock market” task is a prediction task, it would be very useful to know a bit better how subjects learnt to make expectations in this task under placebo and L-DOPA. Could the authors use a canonical RL model (e.g., with a Rescorla-Wagner rule) to estimate the learning parameters of the subjects in the two groups over the course of each block of the task? The authors are proposing in the Discussion that dopamine affects predictions, and it would be very valuable to model the formation of predictions (through basic RL, for example) to know whether and how dopamine (L-DOPA) influences this learning process.

Reviewer #2:

In the present manuscript the authors investigate the role of dopamine in information seeking. Using L-DOPA, which increases dopamine levels, compared to placebo, they test two competing hypotheses about dopamine's involvement in information seeking. The first hypothesis states that increased dopamine leads to increased information seeking, whereas the second hypothesis states that dopamine selectively modulates the impact of valence processing on information seeking. Participants performed a stock market task, in which they could bid to receive or avoid non-instrumental information about their stocks after observing a couple of marked changes (up or down) and rating the expected change to their portfolio and the confidence therein. Echoing previous observations, participants (in the placebo group) were overall more likely to seek information when the market was going up and following larger absolute changes. In line with the second hypothesis, that dopamine selectively modulates valence effects, the authors find no differences in overall information seeking or in the effect of the absolute size of changes on information seeking between L-DOPA and Placebo. Instead, they find that L-DOPA selectively reduces the effect of valence on information seeking.

This work applies a novel approach (pharmacological intervention) to the study of the mechanisms underlying information seeking. This is an area of broad interest and, given the novelty of the experimental manipulation, readers would be interested in it whatever the outcome. I therefore think this could be a valuable contribution to the literature, but I have several concerns about the existing analyses and information provided that need to be addressed.

Major comments:

1) The authors chose a two-step approach when analyzing willingness to pay and choice – each participant's betas from individual regressions were entered into t-tests against zero, as well as independent sample t-tests for comparisons between groups. I would recommend that they instead run a linear mixed effects model, including interactions with treatment group alongside the effects of the market variables. The benefit of this approach would be that all participants can be included and the number of tests to run (with false positive rates increasing accordingly) would be minimized. Further, due to the hierarchical fitting and shrinkage, the obtained parameters would be more robust.

2) For the analysis of the choice data, a number of subjects needed to be excluded because they did not have sufficient observations in either level of the DV. For these data, I would recommend the authors instead just use a generalized linear mixed effects model with a binomial link function to replace the logistic regression (or justify why the current approach is preferable). It also wasn't clear how indifferent trials were handled in this analysis. Were these trials coded as 0 or just excluded? In any case, it would be informative to see how information seeking and avoidance each separately compared to indifference.

3) The authors should clarify the approach to trial-level analysis. As far as I can tell, they aggregated relative frequency of bid to receive vs bid to avoid, (or, in a subsequent analysis, average expectation) within groups and trials and market condition and then analyzed the data with group as intercept and trial and the market condition as predictors. If that's the case, they are effectively generating two artificial participants (placebo and L-dopa) and then assessing the difference between those two, without accounting for between-participant variability. I would recommend instead running a linear mixed effects model across all subjects (collapsing across groups), with subject as random intercept and group as a between subject regressor, and test for interactions between group, trial and the predictors of interest. Given the large number of trials that participants were indifferent, for analyses that focus on the difference between receive vs. avoid, I would recommend reporting whether the groups differed in the number of trials they were indifferent under the same conditions (e.g., market-up vs. market-down) – ideally the proportion of each of these 3 choice types would be visualized at least in the supplement (or reported in a table).

4) The authors interpret their effects in part through the lens of DA influences on encoding of positive vs. negative RPEs. Could they test this more directly by looking at trial history effects (prior outcomes vs. expectations) on subsequent expectations and choices in the two groups?

Reviewer #3:

This study is about the causal effect of changing dopamine in humans indirectly with L-DOPA on information seeking behaviours. To measure information seeking the authors use a decision task in which participants observe a stock market and can decide to pay for information about the consequences on their own personal portfolio. Alternatively, they can be indifferent or even pay to not have to see information. Decisions are made on a sliding scale with increasing money amounts being associated with higher probabilities of the preference being realized. To get additional information about participants internal expectations, they are also asked about their own predictions of portfolio changes and confidence in their prediction.

In the placebo group the authors found a clear preference for paying to see positive information or not paying to avoid positive information (at least early in the task), despite there not being any relevance of this information in increasing or decreasing future rewards. However, participants who were given L-DOPA do not show such a strong valence effect on information seeking behaviour and were instead more driven by the absolute market change (which is related to how much their portfolio might change up or down).

Additionally, L-DOPA increased expectations of portfolio value, but again, this might be more pronounced early in the task.

Overall, the task and data have potential to be interesting to the field. However, certain aspects of the task appear a bit under-analyzed and it is sometimes unclear what the exact effects really are as responses are sometimes highly aggregated and other unusual analytical choices were made. Therefore, it is quite difficult for me to assess what my recommendation would be once those additional analyses and illustrations are made.

Major concerns:

1) One of the most important problems I had with the paper was trying to find out what was driving the effects of L-DOPA. If I didn't misunderstand the behavioural responses, participants could make one out of three choices. A) They could pay to be more likely to see information, B) be indifferent and get info 50% of cases or C) pay to reduce the probability to receive information. Additionally, to this there is an amount they pay to avoid or get information but not when they are indifferent. This means behaviour under L-DOPA can change by either changing the proportions of seek, indifferent or avoid or change the amount participants are willing to pay for each or a mixture of both. However, the authors only present the data in two ways. Either they binarize the data into proportions of seeking vs not (essentially treating indifference like avoidance) or by making a more quantitive measure of willingness to pay (WTP) in which they combine amounts to seek and amounts to avoid by sign-flipping the avoid amounts (it is unclear from the Materials and methods what they do with the indifference trials, but I assume they just ignore them?). From both analyses it is unclear what exactly drives the change they observed. For example, is it a decrease in avoidance or an increase in seeking when the market goes down, that leads to smaller signed effects?

To answer these questions the authors should A) run three logistic regressions, the seeking or not, indifferent or not and avoiding or not (or find another way to look at each choice type) B) they should tease apart the quantitive effects included in the WTP metric further into amounts payed to avoid and amounts payed to seek as these could be very different things. This is important because statements like : "Specifically, administration of LDOPA increased information-seeking about potential losses without impacting information-seeking about potential gains." Rely on the idea that their metric is specifically measuring information seeking, while it is a combination of information seeking, information avoidance and indifference. If I am not mistaken their results could equally well be due to a decreased avoidance behaviours or less indifference (except changed indifference wouldn't obviously affect WTP unless indifference trials were more likely to go into one of the other two categories).

2) It would be more intuitive to show the effect of market change in positive and negative change separately rather than having absolute and signed changes in one regression. That way readers could assess whether both conditions have a modulation by market changes in the control group (because their signed vs unsigned is equally consistent with no effect in negative and a twice as large effect as the absolute in the positive case). I think it is important to know how the regressors add up in this regard because there are quite different interpretations if there is a complete lack of an effect in negative vs increased effect in positive change condition. Including the signed and unsigned regressors this way and testing them against zero is somewhat misleading (if I understood their analysis correctly) because the absolute change regressor can be significant event if their effect only exists in up market trials because of the way a regression adds and subtracts different regressor together and the two regressors are uncorrelated but not independent! This also means this statement is not necessarily true: "Note, that the absence of difference in the impact of absolute market change on information-seeking across groups indicates that both groups encoded the magnitude of the market-change equally well." As the controls might not have encoded the magnitude at all in the negative condition.

3) The way the authors ran the linear mixed effects models seemed a bit strange to me. They write: "We then ran a Linear Mixed Model predicting information seeking exactly as described above (that is the proportion of subjects who selected to seek information minus the proportion who selected to avoid information on each trial)". This suggests the authors did not fit the data for every participant separately but instead just fit the ratio of participant choices across trials separately for each group. If this is what they did, it is quite unusual and there is no rationale for it mentioned in the text. More commonly the model would have been fit on the individual participant level (potentially using a hierarchy). It is also completely unclear why they suddenly introduce a linear mixed effect style model, as they could have used one from the start or analyzed the data without it as they did in the first results they report (logistic and linear regressions). If these analysis choices have a clear rationale, it wasn't obvious from the text. In the Materials and methods again no explanation of why these unusual analysis choices were made but it again sounds like thy used aggregate values across all participants of a group for the fit instead of fitting each participant: "To explore whether the effect of LDOPA emerged over the course of the experiment or appeared from the very beginning, on each of the 200 trials we quantified information seeking as the proportion of participant that paid to receive information minus the proportion who paid to avoid information. We did this separately for each group for trials in which the market was going up and for which the market was going down. Then, we performed a Linear Mixed Model predicting information-seeking with group (LDOPA, Placebo), valence (market up, market down) and time (trial number: 1-200) and their interactions as fixed effect and group as random effect with fixed and random intercept. To further investigate the effect of valence over time we run the same Linear Mixed Model (without valence of course) predicting information seeking separately when market was down and when it was up." This also means that they have to calculate aggregate values for other potentially interesting regressors like expectation "To investigate the effect of LDOPA on expectations we calculated the mean expectation rating of participants on each trial separately for each group when the market was going up and when it was going down. Then we run a Linear Mixed Model predicting expectation rating with group (LDOPA, Placebo), valence (market up, market down) time (trial number: 1-200) and their interactions as fixed effect and group as random effect with fixed and random intercepts." Because expectation isn't used separately for every participant, the aggregate regressor completely neglects to explain any variation between participants that could feasibly explain participants choices.

Furthermore, the linear mixed effects models use the proportion measure, ignoring any magnitude without any reasons for why this analytical choice was made. This feels like throwing out valuable information without giving a reason. If using the magnitude doesn't work the reader should know this, as it is very valuable information.

4) I was quite surprised by one finding later in the paper. "The interaction between valence and group did not reach significance (β = 0.11, F(1,792) = 3.306, p = 0.069), instead there was an interaction between group, valence and time β = 0.001, F(1,792) = 9.450, p = 0.002. As can be observed in Figure 4B the three-way interaction effect was due to LDOPA increasing positive expectations when the market went down more towards the end of the task, but increasing positive expectations when the market was going up more towards the beginning of the task." As this result suggests that there was no maintained valence effect but instead an effect that diminishes over time. While, it is difficult to know whether that is because the drug effects got weaker or other reasons, I think it deserves more attention as the paper currently mostly reads as if there was a straightforward valence effect without changes over time (unless they have other evidence for a non-time dependent valence effect which I missed).

---

## [Author Response]

Main concerns that require revisions:1) Status of information seeking in the task. As emphasized by the authors in the description of their task, information seeking in their “stock market” task is purely non-instrumental. It cannot be leveraged by subjects to gain information that would help them maximizing rewards in the task – something which subjects should be aware of from the task instructions. We agree that information seeking behaviors can be studied in situations where it has no instrumental value. However, it is unclear how one can assume that findings obtained in a condition where information seeking is divorced from reward maximization would generalize to conditions where information seeking has an instrumental value. This is important in this study because the authors study the effect of dopamine on information seeking, and dopamine is also widely known to play an important role in reward-guided learning and decision-making. It is thus important to state, and discuss, the possibility that the results obtained in this study may not apply to tasks where information seeking can be leveraged to maximize rewards.

We completely agree with the reviewers. As we stated in the original Discussion “It is important to keep in mind, however, that our task exclusively examined non-instrumental information about gains and losses. It is possible that dopamine plays a more general role in information-seeking for instrumental information and/or non-valenced information. Future studies are needed to investigate these possibilities.”

One cannot assume that the findings will (or will not) generalize to instrumental information-seeking, which is an intriguing empirical question to be answered with future studies. We now expand on this point in the revised manuscript.

2) Handling of “indifferent” trials and exclusion of subjects. It is unclear to me how the authors dealt with the large number of “indifferent” information seeking ratings provided by the subjects (this is the most populated class of trials in Figure 2). The authors should clarify whether and how this large number of trials was handled in the analyses. We recommend reporting whether the groups differed in the number of trials they were indifferent under the same conditions (e.g., market-up vs. market-down) – ideally the proportion of each of these 3 choice types would be presented at least in a supplementary figure.

Thank you for the opportunity to clarify how trials in which the subject selects to pay 0p (i.e. “indifferent trials”) are handled. In the Willingness to Pay (WTP) analysis these trials are handled as any other trials. We simply insert the number “0” as the dependent measure. This is not a problem, as the dependent measure varies from -99p all the way to +99p. We now make this clearer in the manuscript.

For analysis in which binary “choice” was the dependent measure (e.g., seek information/ not seek information) indifferent trials were categorized as ones in which the subject chooses not to seek information. Following the reviewer comment, however, it became apparent that there are several ways in which the WTP scale can be divided into two categories. We originally binned WTP < = 0 responses as one category and WTP >0 as the other (as explained above). However, one could also divide the data into trials in which the subject chooses to actively avoid information (WTP <0) or not (WTP > = 0). One could also not include zeros at all and bin trials into actively seek information (WTP >0) and actively avoid information (WTP <0). Given that the participants reported their decisions on a continuous scale which provides greater sensitivity, we decided to simply use the continuous WTP scale in all our analysis (as done previously – Charpentier et al., 2019) rather than bin the responses by dividing the scale into two. In this analysis indifference trials are simply coded as zero and thus there is no ambiguity.

Following the reviewers’ recommendation we now report that number of trials in which subjects selected to pay 0p (“indifferent trials”) did not differ between groups under the same condition, and we present these numbers in the supplementary material.

Also, the exclusion of participants based on the empirical variability of their choices appears suboptimal. The authors have adopted a two-step approach when analyzing willingness to pay and choice – each participant's betas from individual regressions were entered into t-tests against zero, as well as independent-sample t-tests for comparisons between groups. We recommend the authors instead run a linear mixed effects model, including interactions with treatment group alongside the effects of the market variables. The benefit of this approach would be that all participants can be included and the number of tests to run would be minimized – thus reducing the type-1 error rate. Further, due to the hierarchical fitting and shrinkage, the obtained parameters would be more robust. As an example, for the analysis of the choice data, a number of subjects needed to be excluded because they did not have sufficient observations in either level of the DV. For these data, we recommend the authors instead use a generalized linear mixed effects model with a binomial link function to replace the logistic regression.

Thank you for this excellent suggestion. We now replaced all our previous analysis with a linear mixed effects model (when examining WTP) exactly as recommended. The analysis reveals the same results as originally reported. For WTP there is a significant interaction between group and valence on the Willingness To Pay for information (*β* = -0.15, CI = -0.29/-0.01, t(230.52) = 2.15, p = 0.032) as well as a main effect of valence (*β* = 0.20, CI = 0.11/0.30, t(229.60) = 4.11, p = 0.0001) and absolute market change (*β* = 0.41, CI = 0.25/0.58, t(231.35) = 4.87, p = 0.0001). There was no interaction between group and absolute market change (*β* = 0.07, CI = -0.17/0.30, t(232.42) = 0.576, p = 0.565), nor a main effect of group (*β* = -2.61, CI = -7.50/2.28, t(232.53) = 1.045, p = 0.297).

While we now focus on the continuous scale and no longer bin responses into two group, we nonetheless conducted the suggested analysis for the purpose of this response and found the same results. When dividing the data into two bins – actively select information (WTP> 0) or not (WTP=< 0) and analyzing the data with logistic Linear Mixed Model we found a significant interaction between group and signed market change (*β* = -0.016, SE = 0.007, z = -2.214, p = 0.027), as well as a main effect of valence (*β* = 0.026, SE = 0.005, z = 4.939, p = 0.0001) and absolute market change (*β* = 0.057, SE = 0.009, z = 6.083, p = 0.0001). There was no interaction between group and absolute market change (*β* = -0.007, SE = 0.013, z = -0.529, p = 0.596), nor a main effect of group (*β* = 0.136, SE = 0.267, z = 0.511, p = 0.609).

3) Fixed-effects analyses in linear mixed models. The reliance on fixed-effects estimation of interaction terms rather than random-effects estimation seems problematic, given that all the key results in the manuscript depend on interaction terms. It appears that the authors did not fit the data for every participant separately but instead just fit the ratio of participant choices across trials separately for each group. If this is what they did, it is quite unusual and there is no rationale for it mentioned in the text. More commonly the model would have been fit on the individual participant level – preferably using a hierarchical approach. It is also completely unclear why they suddenly introduce this type of modeling at this stage. If these analysis choices have a clear rationale, they should be clarified explicitly in the main text since the key results depend on it. However, by aggregating behavioral measures within groups and trials and market condition, and then analyzing the data with group as intercept and trial and the market condition as predictors, the authors are effectively using a fixed-effects approach and thus do not account for between-participant variability. In practice, the authors are generating two artificial participants (placebo and L-dopa) and then assessing the difference between those two, without accounting for between-participant variability. We recommend the authors instead run a linear mixed effects model across all subjects, with subject as random intercept and group as a between-subject regressor, and test for interactions between group, trial and the predictors of interest.

Thank you for this helpful suggestion. We now replaced all our previous analysis with a linear mixed effects model across all subjects, with subject as random intercept and group as a between-subject regressor, and test for interactions between group, trial and the predictors of interest, exactly as recommended. The analysis reveals the same key results as originally reported. Specifically, we find an interaction between market change and group in the loss domain (*β* = 0.40, CI = 0.07/0.74, t (751.900) = 2.35, p = 0.018), but not in the gain domain (*β* =0.10, CI = -0.29/0.50, t (490.600) = 0.505, p = 0.614) on the willingness to pay for information. To tease apart the interaction of interest we run linear mixed models for each group and each valence polarity separately. This revealed that under L-DOPA participants were willing to pay more for information the greater the gains (effect of market change: *β* = 0.79, CI = 0.53/1.05, t (313.300) = 5.98, p = 0.0001) and the greater the losses (effect of market change: *β* = 0.53, CI = 0.33/0.74, t (484.700) = 5.033, p = 0.0001). In contrast, under Placebo participants were willing to pay more for information the greater the gains (effect of market change: *β* = 0.69, CI = 0.40/0.99, t (219.900) = 4.64, p = 0.0001) but did not show this effect for losses (effect of market change: *β* = 0.12, CI = -0.13/0.39, t (335.800) = 0.956, p = 0.339).

While we follow the reviewers’ request and no longer include the analysis looking at proportion of participants selecting information over time, we would like to clarify that we conducted this analysis because it has been previously conducted by us on a similar task (Charpentier et al., 2019 – PNAS, supplementary material, top of page 2 and Figure S4). It was also conducted in the past when looking at how LDOPA alters choice in a reinforcement learning task (e.g., Pessiglione et al., 2006 – Nature, Figure 1B, left panel).

4) Reward history effects to link the results more directly to DA and reward-guided learning. The authors interpret their effects in part through the lens of DA influences on encoding of positive vs. negative RPEs, but this interpretation does not rest on specific analyses of the behavioral data. Because the “stock market” task is a prediction task, it would be very useful to know better how subjects learnt to make expectations in this task under placebo and L-DOPA. For this purpose, the authors could fit a canonical RL model (e.g., with a Rescorla-Wagner rule) to estimate the learning parameters of the subjects in the two groups. They could also (or instead) look at trial history effects (prior outcomes vs. expectations) on subsequent expectations and choices in the two groups. Because the authors are proposing in the Discussion that dopamine affects predictions, it would be very valuable to model the formation of predictions (through basic RL, for example) to know whether and how dopamine (L-DOPA) influences this learning process.

In the Discussion we speculate that DA may influence learning about the affective impact of information. For example, a subject may believe they will not have much of a negative response to knowing they lost but when the information is provided they do have a strong negative reaction. This triggers a negative PE which may influence future information-seeking choices. This negative PE may be dampened by DA. This speculation could be tested in the future by measuring subjects’ predicted affective responses vs actual affective responses.

While affective responses are not record, the reviewer is suggesting that we examine more traditional learning about outcome (i.e. portfolio value). In particular, how the difference between prior portfolio value and prior expectations about portfolio value impact subsequent expectations. This is a reasonable request and indeed DA has been shown to affect this type of traditional learning (Frank et al., 2004, Pessiglione et al. 2006). However, our task was not designed as a learning task of this sort. First, there is no incentive for expectations to be accurate, nor do choices effect outcomes. Second, portfolio values (i.e. “outcomes”) are only observed on half the trials. Third, past outcomes have no bearing on present outcomes. The best way to make a prediction is to simply report the market change on that trial regardless of history. The likelihood that the portfolio will follow the same exact trend as the market is 65% and in 35% it would change in the opposite dircetion with a randomly generated magnitude. Thus, if the market goes up, the most rational response is to expect your portfolio to go up.

Nevertheless, we followed the request and examined if the difference between prior outcomes observed by the subject and prior expectations were relate to subsequent expectations. As detailed below, this did not reveal an effect, which is rationale, as the change in portfolio on the last trial is unrelated to change in portfolio on the next trial. In particular, we run a mixed linear model entering participants’ expectation rating on trial t (note – the rating is always regarding the expected portfolio change relative to the previous trial) as the dependent measure, and as the independent measure the change in outcome on last trial (portfolio_t-1_ minus portfolio_t-2_) minus expectation rating on t-1. We of course only included trials in which portfolio was in fact observed on the last trial. There was no effect (L-DOPA: *β* = -0.001, CI = -0.01/0.008, t (120.787) = 0.315, p = 0.753; Placebo: *β* = 0.001, CI = -0.009/0.12, t (108.300) = 0.304, p = 0.762).As participants often did not observe the portfolio on trial t-2 we ran the analysis again this time instead of inserting portfolio_t-2_ in the equation above we inserted the portfolio value last observed before t-1. Again, this did not reveal an effect (L-DOPA: *β* = -0.001, CI = -0.008/0.006, t (112.877) = 0.330, p = 0.742; Placebo: *β* = -0.0007, CI = -0.006/0.005, t (102.700) = 0.253, p = 0.800). We also examined if past outcome (portfolio_t-1_) was related to current prediction rating, it was not (L-DOPA group: *β* = 0.0001, CI = -0.003/0.002, t (519.000) = 0.464, p = 0.643; Placebo group: *β* = 0.0009, CI = -0.001/0.003, t (213.500) = 0.621, p = 0.535).We then run all these models again, this time predicting WTP for information. As expected none showed a significant effect (all P > 0.240). We now report these results in supplementary material.

5) Implications of significant time-dependent interactions. The fact that it is the three-way interaction between group, valence and time that is significant, not the two-way interaction between group and valence has implications. This result suggests that there was no maintained valence effect but instead an effect that diminishes over time. It is difficult to know whether that is because the drug effects got weaker or other reasons, but in any case it deserves more attention – and explicit discussion – as the paper currently mostly reads as if there was a straightforward valence effect without changes over time. This is of course unless the authors can provide additional evidence for a time-independent valence by group interaction.

Please note that the lack of two-way interaction and existence of three-way interaction is not for the information seeking data, but rather describe the expectations data. Indeed, for participants’ expectations there is no two-way interaction. In fact, when we use the new analysis recommended by the reviewers (mixed linear model) the three-way interaction does not hold. This further supports our conclusion that our key findings regarding information-seeking (that is the significant two-way interaction between group and valence on information-seeking) is not simply a consequence of the effects of L-DOPA on expectations. In the revised manuscript this is now made clear throughout.